

# Bathymetric Properties of the Baltic Sea

Martin Jakobsson[1], Christian Stranne[1], Matt O'Regan[1], Sarah L. Greenwood[1], Bo Gustafsson[2], Christoph Humborg[2], and Elizabeth Weidner[1,3]

[1]Department of Geological Sciences, Stockholm University, Stockholm, 10691, Sweden
[2]The Baltic Sea Centre, Stockholm University, Stockholm, 10691, Sweden
[3]Department of Earth Science, University of New Hampshire, 56 College Road, Durham, NH, USA

*Correspondence to*: Martin Jakobsson (martin.jakobsson@geo.su.se)

**Abstract**

Marine science and engineering commonly require reliable information about seafloor depth (bathymetry), e.g. for studies of ocean circulation, bottom habitats, fishing resources, sediment transport, geohazards and site selection for platforms and cables. Baltic Sea bathymetric properties are analysed here using the using the newly released Digital Bathymetric Model (DBM) by the European Marine Observation and Data Network (EMODnet). The analyses include hypsometry, volume, descriptive depth statistics, and km-scale seafloor ruggedness, i.e. terrain heterogeneity, for the Baltic Sea as a whole as well as for 17 sub-basins defined by the Baltic Marine Environment Protection Commission (HELCOM). We compare the new EMODnet DBM with IOWTOPO, the previously most widely used DBM of the Baltic Sea which has served as the primary gridded bathymetric resource in physical and environmental studies for nearly two decades. The area of deep water exchange between the Bothnian Sea and the Northern Baltic Proper across the Åland Sea is specifically analysed in terms of depths and locations of critical bathymetric sills. The EMODnet DBM provides a bathymetric sill depth of 88 m at the northern side of the Åland Sea and 60 m at the southern side, differing from previously identified sill depths of 100 and 70 m respectively. High-resolution multibeam bathymetry acquired from this deep water exchange path, where vigorous bottom currents interacted with the seafloor, allows us to assess what we are missing in presently available DBMs in terms of physical characterisation and our ability to then interpret seafloor processes and highlights the need for continued work towards complete high-resolution mapping of the Baltic Sea seafloor.

## 1 Introduction

The Baltic Sea's bathymetric properties, including its hypsometry, bottom ruggedness and depths of critical sills, influencing water, nutrient and carbon exchange between the major basins (e.g. Bendtsen et al., 2009;Gogina and Zettler, 2010;Omstedt et al., 2014;Stigebrandt, 2001;Rolff and Elfwing, 2015), internal mixing in deep waters (Lappe and Umlauf, 2016;Nohr and Gustafsson, 2009), and bottom habitats (Kaskela and Kotilainen, 2017), are necessary input parameters to many physical and



environmental studies. Bathymetry is thus often required, preferably compiled into a Digital Bathymetric Model (DBM) suitable for analyses and/or as a framework in numerical models (Hell et al., 2012). A DBM is a Digital Terrain Model (DTM, see Li, 2004) where the terrain specifically represents the seafloor, commonly formatted into a regular grid with depths assigned to the grid cells.

The spatial boundaries of the Baltic Sea are formally defined in the published 3$^{rd}$ edition of the International Hydrographic Organizations (IHO) document S-23 "Limits of oceans and seas" (International Hydrographic Organization, 1953). This definition does not include the Kattegat, the Sound or the Belt Seas (Fig. 1). The 3$^{rd}$ edition of S-23 includes three subdivisions of the Baltic Sea: Gulf of Bothnia, Gulf of Finland and Gulf of Riga. Although not stated in S-23, the water body outside of

these three subdivided areas has commonly been referred as the Baltic Proper. The Baltic Marine Environment Protection Commission (HELCOM), a.k.a. the Helsinki Commission, is an intergovernmental organization formed in 1974 to coordinate and govern actions aimed to protect the environment of the Baltic Sea. HELCOM has implemented a definition of the Baltic Sea that includes the Kattegat, the Sound and the Belt Seas. Furthermore, based on bathymetry and hydrology, HELCOM has defined 17 sub-basins aimed to serve as areas where measured parameters describing the marine environment are to be assessed

and compared regularly (HELCOM, 2018) (Fig. 1).

Here we analyse the Baltic Sea bathymetry using the newly released DBM by the European Marine Observation and Data Network (EMODnet) (EMODnet Bathymetry Consortium, 2018). This DBM has a resolution of 1/16×1/16 arc minutes (~115×115 m) which is substantially higher than previously released by the EMODnet Bathymetry Consortium or other efforts

available to the scientific community (Seifert and Kayser, 1995;Seifert et al., 2001;Hell and Öiås, 2014). We adopt the HELCOM definition of the Baltic Sea and derive geomorphometrical parameters, including hypsometry and descriptive depth statistics, for each of the 17 defined sub-basins as well as for the Baltic Sea as a whole. We additionally explore km-scale seafloor ruggedness, i.e. terrain heterogeneity, across the entire Baltic.

Up until 2014 when the Baltic Sea Bathymetry Database (BSBD) was released (Hell and Öiås, 2014), the most widely used DBM of the Baltic Sea was IOWTOPO, compiled at the at the Leibniz Institute for Baltic Sea Research, Warnemünde (Seifert et al., 2001;Seifert and Kayser, 1995). IOWTOPO provides a grid cell-size of 2×1 arc minutes (longitude×latitude) over the entire Baltic Sea (IOWTOPO2) and double the resolution in the southern region up to 56°30'N (IOWTOPO1). Since IOWTOPO has served as a base for many environmental studies, provided a bathymetric framework in numerical models

(Dargahi et al., 2017;Meier et al., 2003;Tuomi et al., 2018;Lessin et al., 2014) and represented the Baltic Sea in other DBMs covering larger areas of the World oceans (Jakobsson et al., 2008), we include a comparison between IOWTOPO and the newly released EMODnet. The depths and locations of critical bathymetric sills governing deep water exchange between the Bothnian Sea and the Northern Baltic Proper (Fig. 1) are identified and analysed in both DBMs. Furthermore, in the path of this deep water exchange geophysical mapping data are presented from Stockholm University's Research Vessel (RV) *Electra*,



permitting us to assess how meter-scale resolution portrayal of the seafloor bathymetry can improve identification and analyses of seafloor processes. Interaction of past and present currents is clearly visible in the high-resolution mapping data as well as the occurrence of substantial mass wasting. These observations highlight what we are missing in presently available DBMs and the need for continued work towards complete high-resolution mapping of the Baltic Sea seafloor.

## 2 Material and Methods

### 2.1 Digital Bathymetric Models and their sources of error

The resolution of a DBM refers to the size of its grid cells. However, since the depth or height assigned to a grid cell may have resulted from interpolation of source data relatively far from the cell itself, it may be a misleading measure, in particular in the

marine realm where the vast part of the World ocean floor remains unmapped (Mayer et al., 2018). Therefore, information about the underlying source data is required and should be made available along with the release of the DBM. In this work we analyse the EMODnet DBM released 2018 and compare it with the latest update of IOWTOPO1 and 2 from 2008. The EMODnet bathymetric portal provides source references through an online interactive map tool (http://portal.emodnet-bathymetry.eu). This tool makes it possible to investigate the underlying bathymetric sources in any specific area. Furthermore,

EMODnet provides standard deviation of the grid-cell depths where this information has been possible to acquire from the source data. In the Baltic Sea, the standard deviation is mostly assigned a value of 0 m, which is far from realistic and simply reflecting lack of information about the contributed data sources available to the EMODnet compilation team. Error estimations of DBMs based on heterogenic depth data coverages are far from trivial, but can be made if access to the source data and metadata describing data acquisition and associated errors are available. However, even with this information, it is not straight

forward to propagate source data errors to the final depths of the grid cells, which may result from interpolation in the case of sparse source data or subsampling in the case of high data density. Jakobsson et al. (2002) used Monte Carlo simulation to estimate the random error component of an interpolated bathymetric grid by assigning the uncertainties to each contributed source data from information about the navigation and echo sounding systems. The lack of information about the uncertainties associated with the EMODnet grid-cell depths implies that we have to report all results without an estimated uncertainty.

However, the differences we reveal when comparing the DBMs in focus are far beyond any possible associated errors in their underlying data sources, they are of a different magnitude and an effect from largely different underlying source data coverage, which will be discussed.

The IOWTOPO1 and 2 are based on soundings and depth contours, digitized from available bathymetric charts of different

scales, and echo soundings along ship tracks in the deepest parts of the Arkona Basin, the Bornholm Basin, the Stolpe Furrow and the Eastern Gotland Basin (Fig. 1) (Seifert et al., 2001;Seifert and Kayser, 1995). IOWTOPO 1 and 2 include a parameter indicating either the number of original depth values used to derive the mean, minimum and maximum depths in a cell or, if



the cell does not contain any original depth information, the number of neighbouring cells that are used to interpolate a depth. In our comparison, we will mainly make use of the derived mean depths in the cells of both DBMs, although when discussing the deepest location in a given area and bathymetric sills, the deepest depths of grid cells will be used in addition when available.

A more detailed DBM of the entire Baltic Sea bathymetry than IOWTOPO is provided by BSBD (Hell and Öiås, 2014). This DBM has a grid cell-size of 500×500 m on a Lambert Azimuthal Equal Area projection. The compilation work was initiated within a working group of the Baltic Sea Hydrographic Commission (BSHC), consisting of the governmental agencies around the Baltic with hydrographic charting responsibilities. BSBD is based on a significant amount of additional bathymetric source

data compared to IOWTOPO. Hell and Öiås (2014) estimated that between about 30 and 50 % of the Baltic Sea had been mapped to modern standards, primarily using multibeam echo sounders, at the time of the compilation. The spatial coverage of source data is, however, highly heterogeneous and gridding to a much higher resolution than 500×500 m would have been possible in many areas of the Baltic Sea (Hell and Öiås, 2014). An online map tool also exists for BSBD permitting the user to get a view of the source data density, but not the precise origin of the sources (http://data.bshc.pro). The newly released

EMODnet DBM is to a large extent based on the same bathymetric source data as the BSBD, but compiled on a grid with spherical coordinates at the higher resolution of 1/16×1/16 arc minutes (~115×115 m). The Swedish Maritime Administration that led the BSBD compilation work was also responsible for providing the bathymetry in the Swedish waters within EMODnet.

**2.2 Geodetic coordinate reference system and limits of the Baltic Sea**

Before geomorphometric parameters were computed for the EMODnet and IOWTOPO DBMs, the two datasets were projected to Lambert Azimuthal Equal Area projection with the parameters specified in the European Terrestrial Reference System (ETRS) 1989 (ETRS89-LAEA). This geodetic coordinate reference system is recommended by the EU INSPIRE Directive for statistical analyses of data spanning large parts of Europe when true area representations are required (INSPIRE Thematic

Working Group Coordinate Reference Systems and Geographical Grid Systems, 2010). ETRS-89 uses the reference ellipsoid GRS 1980 and the projection parameters are found in most Geographic Information System (GIS) software by searching for the EPSG code 3035. During the projection process, IOWTOPO1 and 2 were combined and resampled to 1000×1000 m and EMODnet was resampled to 115×115 m, i.e. close to the respective DBMs' original grid-cell sizes in geographic spherical coordinates. The resampling and projection of grids as well as the vector processing described below were carried out using

tools available within QGIS, version 3.4.2-Madeira (QGIS Development Team, 2018).





Polygons delineating the Baltic Sea and the 17 defined sub-basins were downloaded in shape-file format from HELCOM. These were dissolved so that only one outer boundary remained for each individual sub-basin as well as for the polygon representing the entire Baltic Sea, i.e. all islands were removed (Fig. 1). The polygons representing the Baltic Sea and all 17 sub-basins were subsequently simplified using the Douglas-Peucker algorithm so that the minimum spacing between the nodes

was left to be equal to or higher than 100 m. The simplified polygons were used in all geomorphometric calculations to constrain them to the HELCOM-defined Baltic Sea or any of its 17 sub-basins.

### 2.3 Geomorphometry

Geomorphometry is the field of quantitative analyses aimed to describe and characterize the Earth's surface terrain (Pike et

al., 2008). It commonly involves analyses of DTMs using GIS software. Hypsometry is a widely used geomorphometric parameter referring to measured heights or depths relative to sea level. A hypsometric curve displays the area distribution as a function of height or depth within a given geographic region. The tool "Hypsometric curves" in QGIS was used to derive hypsometric curves for the EMODnet and combined IOWTOPO 1 and 2 DBMs in all 17 sub-basins as well as for the entire Baltic Sea as one region. Area calculations were made in QGIS at 1 m depth intervals planimetrically on the ETRS89-LAEA

coordinate reference system. Descriptive statistics on the mean depths provided by the DBMs were calculated using the "Zonal Statistics" tool in QGIS. It should be noted that the reported maximum depths in each sub-basin are a "mean" maximum depth from a specific grid cell since the EMODnet does not report max values for each grid cell as some data contributors only provided mean depths.

A quantitative measure of seafloor ruggedness, sometime referred to as roughness, can be computed using several different methods (e.g. Wilson et al., 2007;Pike et al., 2008). Here we calculate Terrain Ruggedness Index (TRI) using the algorithm available within the Open Source SAGA (System for Automated Geoscientific Analyses) tools (Conrad et al., 2015). TRI provides a measure of the bathymetric/topographic variation around a central pixel (Riley et al., 1999). The sum of the absolute differences between the neighbouring cells and the centre cell is averaged. For 3×3 grid cells this follows:

$$TRI = \frac{\Sigma|x_{ij}-x_{00}|}{8}$$   Eq. 1

where $x_{ij}$ = depth of each neighbouring cell relative to the centre cell $x_{0,0}$. The result is scale-dependent, i.e. dependent on the grid-cell resolution of the analysed DTM. For this reason, it is common to vary the size of the region over which the terrain is analysed, i.e. the 'neighbourhood', depending on whether the study is concerned with local or regional variations. In Eq. 1 this is simply done by increasing the block of grid cells for which sum of the absolute differences are compared to the central

cell. Our study aims to provide a regional basin scale perspective.  A radius of 10 grid cells (1000 m), yielding a total block size of 2100×2100 m, was decided after trials to provide interpretable results.



### 2.4 Bathymetric sills

Locations of bathymetric sills, i.e. the deepest depth of a generally shallow zone that would otherwise hinder transfer of water and sediment between two basins, were mapped and analysed using the software Fledermaus by QPS and QGIS. The sills were first identified in the EMODnet DBM. Bathymetric profiles perpendicular to the sills were then generated and compared to
profiles between the same points generated from the IOW bathymetry.

### 2.5 Geophysical Mapping with RV *Electra*

Expedition EL17-IGV04 with RV *Electra* carried out marine geophysical mapping within a focused area in the Southern Quark between Sweden and Åland from Aug 6 to 17, 2017 (Fig. 1). The complete field work included geological coring,
oceanographic stations, in situ sediment temperature logging and geophysical mapping including multibeam bathymetry, sub-bottom profiling, and midwater sonar. We briefly describe the acquisition methods of the data presented in this work, i.e. the multibeam bathymetry and midwater imagery. RV *Electra* has a Kongsberg EM2040 0.4°×0.7°, 200-400 kHz, multibeam echo-sounder and a Kongsberg EK80 wide-band split-beam sonar for midwater mapping operating at two frequencies (70 kHz and 200 kHz). The multibeam is operated using Kongsberg's Seafloor Information System (SIS), version 4.3.2 (Build 31,
DBVersion 30.0) while the split-beam sonar is operated using Kongsberg's dedicated software, EK80 version 1.8.3. Both systems receive position, heading and attitude data from a Kongsberg-Seatex Seapath 330+ navigation unit with the MRU5+ motion and reference sensor. The system is dual frequency (L1/L2 band) and capable of using both GPS and GLONASS satellites. Real-Time Kinematic (RTK) corrections, were received from SWEPOS (https://swepos.lantmateriet.se/) over the internet. This resulted in a horizontal accuracy generally below 5 cm and and a slightly coarser vertical accuracy. Post
processing of the multibeam bathymetry was done using QIMERA software by QPS, version 1.7.2, and midwater images from the EK80 data were compiled using Matlab routines.

## 3 Results

### 3.1 Geomorphometry

By comparing the hypsometric curves of two different DBMs of the same region, differences in specific depth intervals can readily be identified as well as systematic biases in the bathymetric source data. Smith and Sandwell (1997) showed that a bias toward gridded digitized depth contours could be seen as spikes in the hypsometric curve of ETOPO-5, the first global gridded compilation of the World ocean (National Geophysical Data Center, 1988). Biases toward 10 and 5 m intervals are clearly seen in the IOWTOPO hypsometric curve representing the entire Baltic Sea, specifically pronounced between 100 and 50 m
(Fig. 2a). Here the biases are clearly an effect of depths being sampled from charts in steps of 10, 5 and 1 m within the depth



intervals deeper than 150 m, 150-50 m, and 50-0 m respectively (Seifert and Kayser, 1995). A clearly visible difference between the IOWTOPO and EMODnet bathymetries is identified at depths shallower than ~15 m where IOWTOPO has much larger shallow areas (Fig. 2a). Apart from this difference and the spikes, the hypsometric curves of the two DBMs are rather similar, however with a more persistent deviation between 40 and 25 m.

The analyses of the 17 sub-basins show that differences in depths shallower than ~15 m are less apparent in Kattegat, Kiel Bay, Gdansk Basin, and Eastern Gotland Basin (Figs. 2b,e,i,j). Spikes related to biased sampling of data at 10 and 5 m intervals in the IOWTOPO DBM are more visible in Bornholm Basin, Eastern Gotland Basin and Bothnian Sea (Figs. 2h,j,p). The largest differences in the hypsometry are apparent for the Gulf of Finland and the Quark (Figs. 2n and q).

The overall shallower character of the IOWTOPO DBM is apparent in the descriptive statistics. The median/mean depths of IOWTOPO and EMODnet within the Baltic Sea limits are 39/50 m and 42/53 m respectively (Fig. 2, Table 1). All sub-basins have deeper median and mean depths in EMODnet, except for Kattegat and Eastern Gotland Basin (Fig. 3, Table 1). Eastern Gotland Basin has the deepest median and mean in IOWTOPO, while this is instead the case for Northern Baltic Proper in

EMODnet. Across all metrics plotted in Fig. 3, the greatest deviation between the IOWTOPO and EMODnet DBMs is seen in the Western Gotland Basin, Northern Baltic Proper, Åland Sea and the Quark. These cases cover shallow, moderate and deep basins, indicating dataset deviation at all depths on a sub-basin-scale. Åland Sea has the largest difference in median and mean depths between IOWTOPO to EMODnet, (median/mean) 7/26 m compared to 19/37 m (Fig. 3, Table 1).

The area of the Baltic Sea, calculated by summarizing all grid cells of the EMODnet DBM falling within the HELCOM defined boundary and with values of $\leq 0$ m, is ~417×10$^3$ km$^2$ (417,115 km$^2$) (Table 1). The area comes out ~0.2 % smaller when summarizing the separate areas of the sub-basins due to "loss" of grid cells because the QGIS routine only counts complete cells falling within the polygon boundaries. Furthermore, it should be noted that counting grid cells yields a combined area of all the sub-basins that is ~0.1 % smaller than when the HELCOM original polygons of the sub-basins are used to calculate the

area of the Baltic Sea. The reason for this is likely the same, i.e. grid cells only partially within the delimiting polygons are omitted. However, these differences are very small and since this study aims to analyse the DBMs, we have counted grid cells on the Lambert Equal Area Projection for all area calculations. The coarser IOWTOPO DBM has an area of ~427 km$^2$ (427,470 km$^2$), which is as much as ~2.5 % larger than EMODnet. This can most likely be explained by the coarser resolution and all islands that are left out. The volume of the Baltic Sea is ~21.9×10$^3$ km$^3$ (21971 km$^3$) using EMODnet and ~21.2×10$^3$ km$^3$

(21258 km$^3$) using IOWTOPO, i.e. the latter yields a ~3.1 % smaller volume (Table 1). This can be explained by the shallow bias in IOWTOPO seen in the hypsometry.

On a basin scale, the ruggedness of the seafloor is shown to vary spatially (Fig. 4). Particularly rugged areas are confined to the eastern part of the Kattegat, northern Western Gotland Basin from about 57°25'N, northern two-thirds of Baltic Proper



(highest TRI values near the Swedish coast), the entire Åland Sea and north-western sectors of Bothnian Sea and Bothnian Bay respectively. In the Åland Sea, the rugged seafloor is clearly confined to a pattern of rather straight channels, sometimes crisscrossing each other (Fig. 4a), while in other areas the rugged seafloor shows a sinuous pattern. An example of the latter is the band of rugged seafloor stretching from the lower western corner of the Baltic Proper to about 59°40'N 26°40'E in the
southern Gulf of Finland (Fig. 4b). A qualitatively similar band of less pronounced sinuosity is apparent in the northern Eastern Gotland Basin. South of here, the much less rugged nature of the southern Baltic Sea is clearly apparent in the TRI map (Fig. 4). We discuss below possible sources of seafloor ruggedness, but note here that inconsistencies in the bathymetric source data coverage are readily apparent in the TRI map, where roughness 'borders' are unnaturally straight and clearly delineate input data with different native resolutions (Fig. 4c). This highlights the caution needed when interpreting a DBM compiled from
heterogeneous source data, something that will be further addressed in the discussion.

### 3.2 Bathymetric sills

The Åland Sea separates the Bothnian Sea and the Northern Baltic Proper (Fig. 1). The seafloor bathymetry is highly complex: it is a broad zone, much of it shallow, but deep incisions cut through what otherwise could act as an effective barrier to water
exchange. A deep basin is located west of the islands of Åland, with water depths exceeding 200 m over much of its central part (Fig. 5a). Therefore, the threshold that may influence deep water exchange between the Northern Baltic Sea Proper and Bothnian Sea must be searched for both north and south of this deep basin. We analyse the mean depths provided by the EMODnet 2018 DBM because in this area, there are no maximum depths provided or, more precisely, those provided are the same as the mean. The southernmost sill depth (profile W-W´ in Fig. 5) is ~60 m deep and located at the southern end of a
nearly 40 km long, ~1-2 km wide, winding channel that ends in the north in a small E-W elongated basin south of Lågskär (hereafter referred to as Lågskär Basin) with depths exceeding 150 m (Fig. 5a). South of the southernmost identified sill (W), the bathymetry is complex and there are a few points that also may act as sills as they are just about deeper than 60 m. At the northern end of Lågskär Basin there are three sills slightly deeper than 60 m, one in the east and two in the west separated by approximately 15 km and all situated at about the same latitude as Lågskär (profile V-V´ in Fig. 5). North of the deep main
basin of the Åland Sea, a more than 45 km long channel, extending in an almost south-north direction has the shallowest point just about reaching 88 m (profile U-U´ in Fig. 5). Our analyses of the mean depths provided by the EMODnet DBM suggest that transport from/to the Baltic Proper is limited by a ~60 m sill south of the Åland Sea, while transport from/to the Bothnian Sea is limited by a ~88 m threshold north of Åland (Fig. 5a).

The mean depths of the IOWTOPO DBM naturally provide a much more generalized portrayal of the seafloor morphology due to the substantially lower resolution. While the pronounced bedrock channels are not as visible in the IOWTOPO bathymetry as they are in EMODnet, the two main channels where the sills were found are (Fig. 5b). A southernmost sill occurs nearly at the same location in both datasets, although the mean depth in IOWTOPO lies at 49 m instead of 60 m.



Towards the main deep basin there is only one distinct deep passage of about 49 m instead of the three 60 m passages identified in EMODnet. The northern sill is located nearly at the same place as in EMODnet, although its mean depth is 57 m which is substantially shallower. IOWTOPO does, however, provide information on the deepest depth in the cells in this region. At the southernmost sill (profile W-W') the deepest depth is deeper than 80 m, in fact similar to the deepest depth at the northernmost

sill (profile U-U', Fig. 5c). The much coarser resolution of the IOWTOPO gives a shallow bias to the *mean* depths for the thalweg of a channel that is nearly as narrow as the grid-cell size.

### 3.2 High-resolution bathymetry in the Southern Quark

The Swedish Maritime Administration mapped large areas of the Southern Quark using multibeam, and provided subsampled

multibeam grids to the compilation of the EMODnet DBM. The IOWTOPO, on the other hand, is both of substantially lower resolution and based on gridding sparse digitized soundings. For analysis of the effects of resolution downgrading for DBM compilation, we compare the DBMs with multibeam bathymetry acquired by RV *Electra* in the Southern Quark, gridded at a grid-cell size of 2.5×2.5 m (Figs. 6 and 7). The first order comparison shown in Figure 6 reveals the immense difference with IOWTOPO failing to capture the distinct ~2 km wide western channel and the two major ridges protruding east of the channel

as well as the ~1 km wide passage between them. The bathymetric profiles between X-X' and Y-Y', crossing the narrow main western channel, show that the EMODnet DBM portrays the main morphology rather well compared to the high-resolution RV *Electra* surface, while IOWTOPO differs in depth by as much as 100 m in places.

A closer inspection of the RV *Electra* multibeam bathymetry shows a dynamic local environment at the seafloor with, for

example, visible erosional channels, mass wasting, and a sediment drift deposit (Fig. 7). There are hints of some parts of the channels and the drift deposit in the EMODnet DBM, but without knowing where to look from the higher resolution information, most features would not be possible to identify. This shows that there is still enormous value in 'full' resolution data for identification and interpretation of features that either drive or are a product of process interaction between seabed and overlying water column. While the wreck from the 90 m long ship August Thyssen (sunk in 1940 after hitting a mine) ~55 m

water depth is visible in the 2.5×2.5 m, a higher resolution rendition using the full multibeam information shows that there is substantially more information in the acquired multibeam bathymetry than revealed by a 2.5×2.5 m grid (Fig. 7b). There are, for obvious reasons, no signs of August Thyssen in the EMODnet DBM.

### 3.3 Water column imagery

Mid-water acoustic profiles were collected along a part of transect X-X' and the entire Y-Y' (Figs. 6 and 8). Acoustic data can be used to observe features within the water column in a similar manner as sub-bottom profilers or seismic reflections systems



are able to identify geological layers within the stratigraphy below the seafloor (Jakobsson et al., 2016b). Acoustic impedance contrasts, caused by changes in water sound velocity and density, cause reflections and scattering of the acoustic signal. Scattering from point sources as well as reflections from laterally extended acoustic impedance contrasts are clearly visible in both profiles (Fig. 8). The strongest point echoes occur in water depths between about 75 and 100 m. There is a scattering

layer below 100 m in transect X-X' (Fig. 8a) and a section of less coherent, but pronounced, reflections above the bathymetric peak in transect Y-Y' formed by the wreck (Fig. 8b).

## 4 Discussion

### 4.1 Basin-scale morphology and DBM evaluation

The Baltic Sea's bottom topography, hypsometry and depths of critical sills between the major basins have been described by Leppäranta and Myrberg (2009). Their description builds in turn on the published bathymetric characterization of the Baltic Sea by Fonselius (1995). Both these studies were based on analyses of traditional bathymetric maps with depth contours. The first compiled DBM encompassing the entire Baltic Sea was IOWTOPO 2 published in 1995 (Seifert and Kayser, 1995). While this DBM served as the primary resource for gridded Baltic bathymetry for nearly two decades until BSBD was released in

2014 (Hell and Öiås, 2014), it has to our knowledge not been subjected to similar bathymetric analyses as those made by Fonselius (1995) and Leppäranta and Myrberg (2009). Our study does not aim to fully replicate their seafloor analyses using the new EMODnet DBM. First, it would require that the exact same definitions of all sub-basins are applied. Second, we find it more useful to focus on comparing EMODnet with IOWTOPO and the characteristics of these two DBMs because modern uses of seafloor bathymetry rely almost exclusively on gridded bathymetric models. The decision to instead use the HELCOM

definitions of the Baltic Sea and its sub-basins is justified as they are becoming standard in modern assessments of the marine environmental conditions (http://www.helcom.fi/baltic-sea-trends ). Nonetheless, the calculated area and volume for the entire Baltic Sea can still be compared directly to previous studies because only the sub-basins are defined differently in HELCOM.

Leppäranta and Myrberg (2009) reported an area and volume of the Baltic Sea (including the Kattegat) of 415265 km$^2$ and

21720 km$^3$, respectively, compared to our result of 417115 km$^2$ and 21971 km$^3$ based on EMODnet. The area and volume are thus ~0.45 % and ~1.16 % larger for EMODnet, which must be considered rather close matches since the two base datasets are different, both with respect to age and type (contour maps versus DBM). If we instead compare calculated area and volume between EMODnet and IOWTOPO the differences are in fact much larger. IOWTOPO yields an area ~2.5 % larger and a volume 3.1 % smaller than EMODnet. The explanation for these differences is found in the hypsometric curves (Fig. 2). The

fact that IOWTOPO shows a larger area represented by depths shallower than 15 m, particularly noticeable in the depth range of a couple of meters, will affect the volume by making it smaller. This shallow depth bias along coasts and islands is simply due to the coarser grid-cell resolution. There are thousands of islands and small slivers of land along complex coast lines that



not are resolved and instead assigned a shallow depth during the interpolation process. A further consequence of this effect is that the ocean area increases overall.

We expect that future calculations of area and volume of the entire Baltic Sea based on a further improved DBM, will yield
only minor differences compared to the numbers presented here. The Baltic Sea mean depth of 53 m, calculated from the mean depth values in the grid cells of the EMODnet DBM, is within the 53-55 m that is commonly stated in encyclopaedias and published literature, although often without references to the used bathymetric dataset or applied definition of the Baltic Sea. However, there are other depth related parameters that are more sensitive, for example the location of critical sills where a lack of bathymetric source data in small regions may have large effects (see discussion below).

### 4.2 Seafloor ruggedness

Ruggedness might be relevant to a number of geoscientific fields as a heterogeneous seafloor, for example: represents an aspect of 'geodiversity', with implications for habitats (biodiversity) (Kaskela and Kotilainen, 2017); has implications for mixing and stratification (Umlauf et al., 2018;Jayne et al., 2015); influences flow over the seafloor, both now (bottom currents, sediment
transport) and in the past (ice flow, glacial erosion and sediment transport, ice flow stability) (Kietzig et al., 2009). On an ocean-wide scale, for instance, the vertical mixing that occurs over rough sections of the Mid-Atlantic Ridge and other topographically complex areas in the world oceans, influences the global overturning circulation (Wunsch and Ferrari, 2004;Ledwell et al., 2000).

The generalized bedrock geology map of the Baltic Sea by Uścinowicz (2014) reveals that TRI values and patterns in EMODnet coincide with variation in bedrock composition and positions of major faults and structures. The criss-crossing pattern of high TRI values along the Swedish east coast, beginning at northern Öland and stretching across the Åland Sea and along southern Finland (Fig. 4), coincides with the predominance of Proterozoic crystalline bedrock of the Baltic Shield. High TRI values along nearly straight lines follow major faults. The similar TRI-pattern distinguished in Kattegat and from about
62°30'N in the Bothnian Sea, albeit with less pronounced criss-crossing and straight lines of high TRI values, also occur in areas generally composed of Proterozoic crystalline rocks. The sinuous pattern of high TRI values extending from the lower western corner of the Baltic Proper and further into the southern Gulf of Finland (Fig. 4b), occurs where the generalized geological map by Uścinowicz (2014) shows a narrow belt of Cambrian sedimentary rocks, mainly sandstones. South of this belt, Ordovician, Silurian and Devonian clastic and calcareous rocks provide the foundation for a smoother seafloor, which is
reflected in the TRI-map (Fig. 4). Ancient crystalline surfaces have undergone (extremely) long periods of weathering and erosion, and fracture/joint patterns have been exploited to give a very visible surface morphological expression. This contrasts with horizontally, or slightly inclined, bedded sedimentary strata where surface expression is only really apparent when bedding planes crop out.





The Baltic Sea's bedrock geology is mainly inferred from seismic reflection and refraction surveys, dredging, and drilling (Grigelis, 2011). The apparent correlation between bedrock type and seafloor ruggedness suggests that a high resolution regional DBM could be a significant help to further refine bedrock boundaries and for discovering outcrops. However, the

TRI-value pattern is not all inherited from the bedrock. We have already noted its dependence on the size of analysis neighbourhood and the effect of heterogeneous input datasets, where high-resolution survey data have been down-sampled. Furthermore, the macro-scale bedrock topography of the Baltic is overlain by Quaternary glacial, post-glacial and modern sediments and landforms, whose local-scale morphology is superimposed on the underlying relief. High TRI values with a locally very patchy distribution in Figure 4c reflect a drumlin field (Greenwood et al., 2017) superimposed on an otherwise

rather low relief surface. Here the boundaries of the multibeam dataset that reveals these drumlins is also clearly seen in the TRI-pattern highlighting the need to consider the underlying source data when interpreting seafloor morphology using DBMs. This is further emphasized in our analyses of bathymetric sills.

### 4.3 Bathymetric sills and seafloor processes

Bathymetric sills in the Baltic Sea have been much discussed within the oceanographic community because of their influences on circulation patterns and direct control on water exchanges between basins and mixing (e.g. Laanearu and Lundberg, 2000;Lass and Mohrholz, 2003;Gustafsson, 2000;Omstedt et al., 2014). The sills affecting deep water exchange between the Bothnian Sea and the Northern Baltic Proper across Åland Sea (Ehlin and Ambjörn, 1977) will be discussed here because we have a high-resolution perspective provided by the RV *Electra* survey of a section of the overflow area. In contrast to the well

described and investigated sills and thresholds in the Danish sounds, the exchange of water between the central parts of the Baltic Sea and the Bothnian Sea is relatively unknown, especially the northbound flow of salt water and nutrients, which has been suggested to trigger major ecosystem changes in the Bothnian Sea both at present (Rolff and Elfwing, 2015) and in past (Jilbert et al., 2015). Leppäranta and Myrberg (2009) identified three bathymetric sills influencing deep water exchange across Åland Sea: Southern Quark Strait (100 m), between Söderarm and Lågskär (70 m), and in a narrow channel in southern Åland

Sea (70 m). These are three locations where we also locate the critical bathymetric sills in EMODnet, but find them all to be shallower: 88 m in the Southern Quark Strait and 60 m in the two other locations (Fig. 5a). In this context, it is appropriate to discuss the fact that depths provided by a DBM such as EMODnet represent grid cells, in our analysis having a size of ~115×115 m. EMODnet only contains mean depths for the grid cells in this particular region and no maximum or minimum depths, because the underlying source data from the Swedish Maritime Administration lacks this information here. When a

maximum depth is provided for a grid cell, it could be used as the depth of a sill. However this may be misleading because the maximum depth could be surrounded by shallower depths from the grid-cell area. If this is the case, the maximum depth would instead represent a local depression. The opposite is true if the minimum depth is used as it could be from a local obstacle. Neither will the mean depth always be the most representative of a sill as large depth variations within the grid-cell area may





exist. The problem of selecting the right depth increases with lower resolution DBMs, which is clearly illustrated by our analysis of IOWTOPO 2 (Figs. 5b and c). The coarse resolution of the grid cells (originally 2×2 arc minutes) makes it impossible to capture the critical details in the region between the Bothnian Sea and the Northern Baltic Proper where the sills are located (Fig. 5b). The problem is greater if the DBM is based on a sparse underlying source dataset requiring interpolation.

Even if available ocean circulation models are not able to make use of the resolution provided by EMODnet, except when applied over small areas, the sub-sampling from higher to lower resolution can be made in such a way that critical sills are preserved.

The comparison between IOWTOPO 2, EMODnet and the multibeam data from RV *Electra* in the Southern Quark area shows

the strength of compiling a DBM by sub-sampling full coverage high-resolution bathymetry instead of interpolating from heterogenic and sparse single beam depth soundings (Fig. 6). The fact that EMODnet in the Southern Quark is based on complete multibeam surveys results in the main seafloor features being well portrayed, although the steepness of the walls and peaks of ridges are lost when downgrading the original resolution (Fig. 6d). From this it becomes clear that in critical areas, such as where bathymetric sills govern water circulation, full multibeam surveys are required for appropriate representation of

the bathymetry, but that it may be adequate at a downgraded resolution.

For further insight into local-scale seafloor processes, full resolution multibeam bathymetry provides valuable additional information (Fig. 7). While a full description of the seafloor features in the mapped area is beyond the scope of this paper, we point out and discuss some visible characteristic bedforms indicative of past glacial activity, bottom currents and mass wasting.

Glacial landforms are common in the surveyed area. For example, there is a semi-circular ~70 m wide and ~4-5 m deep pit with pushed up rims near the mapped wreck of August Thyssen (Fig. 7b). Similar features are widespread further north on crests of drumlins mapped by multibeam (Greenwood et al., 2017;Jakobsson et al., 2016a). The pits are interpreted to form when icebergs lose their balance, due to melting or partial disintegration, and rotate to temporarily reach deeper with one corner making a dent in the seafloor or, alternatively from icebergs with expressed pointy keels that ground in calm conditions

so that they lift of the seafloor before elongated scours are formed. The image in Figure 7b showing the iceberg pit and August Thyssen is created from a 50×50 cm grid from a specific survey over the wreck using 400 kHz mode instead of 300 kHz.

Characteristic seafloor bedforms, both erosional and depositional, have long been used to provide information on bottom current velocity and flow direction in studies of both modern and past oceanographic conditions (Hollister and Heezen,

1972;Kenyon and Belderson, 1973). With high-resolution multibeam bathymetry acquired with surface vessels in relatively shallow waters such as in the Baltic Sea, we are able to make use of bedforms scaling from a few decimetres in size at the very best or, more commonly, from a few meters. Our multibeam bathymetry shows bedforms indicating substantial bottom currents in several areas of the Southern Quark. For example, along the western and northern foot of the steep wall of the ridge where mass wasting occurred, bottom currents appear to have scoured a >20 m deep and >200 m wide channel (Fig. 7c). A similar





but smaller erosional channel is visible along the northern foot of the twin-like ridge located to the west. Next to the channels, the smooth texture and rounded seafloor morphology suggests sediment accumulations that may be drift deposits. These would be typical targets for further geophysical surveys and coring since they may contain a high-resolution sedimentary record of the bottom current flow over time across Åland Sea. Stow et al. (2009) constructed a bedform-velocity matrix that permits a

first-order inference of bottom current velocity from mapped bedforms. This bedform index includes elongated erosive features around obstacles, often with elongated erosive tails. We identify these kind of bedforms, often called obstacles with comet marks, in the northern part of the surveyed area (Fig. 4d). The directions of their tails indicate a prevailing bottom current flow towards south-southeast. These bedforms, when large, may form under prevailing current flow regimes with velocities >1 m/s (Stow et al., 2009). Key to this matrix is sediment composition implying that this information must supplement the shape and

size of the bedforms inferred from geophysical seafloor mapping.

### 4.4 Adding the midwater perspective

Midwater echo sounders permit remote observations of thermohaline stratification (Stranne et al., 2017), turbulence (Farmer and Dungan Smith, 1980;Moum et al., 2003), suspended particles (Young et al., 1982;Hay and Sheng, 1992), as well as

individual fish, fish schools, and zoo plankton (Chu et al., 1994). Advantages of the new type of wideband echo sounders that we used in this study compared to conventional narrow-band systems include increased signal-to-noise ratio and increased range resolution (Stanton and Chu, 2008), as well as the ability to study the frequency response of individual targets to help identify the source of the acoustic backscatter (Weidner et al., 2019;Irish et al., 2010). While this kind of frequency response analysis has not been done on the data presented here, we can still visually identify some specific features in the acoustic mid-

water profiles such as fish schools, zooplankton/suspended particles, thermohaline stratification (verified with co-located CTD data) and turbulence (Fig. 8). It is clear that the dramatic and steep bathymetric features in the Southern Quark influence, in some cases likely even cause, processes in the water column (Fig. 8). This shows that through the combination of high resolution bathymetry data and new wideband sonar technology, we can now collect acoustic data during surveys that will allow us to link (and possibly quantify) vertical mixing within the ocean interior associated with specific bathymetric features.

In the Baltic Sea, mixing inferred from direct observations is typically one order of magnitude smaller than when quantifications of mixing are made from measured salinity variance (Reissmann et al., 2009). Although some of the "missing mixing" is likely related to upwelling and double diffusion (Umlauf et al., 2018), local mixing associated with rough and steep bathymetry might be underestimated in the Baltic Sea. This opens up for future studies where seafloor ruggedness can serve as a first order indication of where midwater echo sounding surveys combined with oceanographic stations could provide a

better and more complete view of mixing processes in the Baltic Sea.

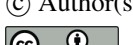



## 5 Conclusions

Comparison between the IOWTOPO and EMODnet hypsometries shows that the area shallower than ~15 m is overrepresented in IOWTOPO over much of the Baltic Sea, while depth differences between the two DBMs otherwise occur at various depth intervals in the different HELCOM sub-basins. This general shallow bias in IOWTOPO is mainly an effect from its coarser

5   resolution. The shallow bias is also evident in the median and mean depths calculated for the two DBMs (IOWTOPO: median=39 m; mean 50 m; EMODnet: median=42 m; mean 53 m). The Baltic Sea area, defined as where grid-cell values are ≤ 0 m in the EMODnet DBM within the HELCOM spatial limits of the Baltic Sea, is ~417×10³ km² (417,115 km²) and the volume is ~21.9×10³ km³ (21971 km³). Using IOWTOPO, the calculated area is ~2.5 % larger while the volume is ~3.1 % smaller.

Analysis of km-scale seafloor heterogeneity, through calculation of Terrain Ruggedness Index (TRI) values using the EMODnet DBM, reveals patterns that generally coincide with variation in bedrock composition of the Baltic seafloor and positions of major faults and structures, with deviations where prominent glacial landforms, e.g. drumlin fields, superimpose the underlying relief. TRI-patterns originating from heterogenic bathymetric source data are also evident from the analysis.

Three areas having bathymetric sills likely influencing deep-water exchange across the Åland Sea, are identified in the EMODnet DBM: 1) In Southern Quark Strait (sill depth: ~88 m, at about 60°26.6'N 18°56.8'E), 2) at three locations along a transect from north of Söderarm to east of Lågskär (sill depth at all three: ~60 m), and 3) in a narrow channel in the Northern Baltic Proper (sill depth: ~60 m at about 59°30.1'N 20°37.3'E). The locations of these bathymetric sills have previously been

20   identified, although their depths were assumed to be significantly deeper. The IOWTOPO DBM suggest both different locations and depths of bathymetric sills that would influence water exchange across the Åland Sea, which is an effect of its lower resolution and less bathymetric source data available during the compilation.

High-resolution multibeam bathymetry from the Southern Quark shows that the EMODnet DBM, here based on downgraded

25   multibeam bathymetry, captures the general topography rather well but fails to reveal mass wasting, seafloor features indicative of bottom currents, and glacial landforms evident in the high-resolution bathymetry. This shows the enormous value in 'full' resolution bathymetric information in marine research and the need for a complete high-resolution mapping of the Baltic Sea seafloor.



**Data availability**

The EMODnet DBM is available for download from the portal: http://portal.emodnet-bathymetry.eu .The IOWTOPO is available from the Leibniz Institute for Baltic Sea Research Warnemünde at https://www.io-warnemuende.de/topography-of-the-baltic-sea.html. The multibeam bathymetry and midwater imagery acquired by RV *Electra* presented in this work have

been granted public release by the Swedish Maritime Administration (release 17-03187). These data are available for download from the Bolin Centre Database: https://bolin.su.se/data/ .

**Author contribution**

M. Jakobsson prepared the manuscript with input from all co-authors. M. Jakobsson analysed the DBMs and processed the RV *Electra* multibeam bathymetry and C. Stranne processed the EK80 data. M. Jakobsson, M. O'Regan, and L. Weidner

carried out the geophysical mapping with RV *Electra*.

**Author contribution**

M. Jakobsson participated in the compilation of the EMODnet DBM, but we do not find that this should be considered a

competing interest. No other authors find any competing interest to declare.

**Acknowledgement**

We thank the crew and Captain of RV *Electra* and the Baltic Sea Centre for their support. The survey with RV *Electra* forms a part of a project financed by the Swedish Radiation Safety Authority. M. Jakobsson worked on this paper during sabbatical

leave supported by Stockholm University and thank NIWA (National Institute of Water and Atmospheric Research) in Wellington, New Zealand, for providing a work space during the sabbatical. Dr Geoffroy Lamarche at NIWA is specifically thanked for fruitful discussions.





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

**Table 1:** Descriptive statistics of the EMODnet and IOWTOPO DBMs calculated using the depths of the grid cells in respective DBMs. Note that the grid cells themselves give the mean of soundings within that cell. For example, the maximum depth in the Western Gotland Basin corresponds to the Landsort Deep, commonly cited as 459 m, yet the corresponding cell in EMODnet gives 454 m and in IOWTOPO is even shallower (402 m), because the coarser resolution implies that grid cells contain mean depths derived from larger areas.

**Figure 2:** Bathymetric map of the Baltic Sea based on the EMODnet 2018 DBM. The HELCOM-adopted outer limit of the Baltic Sea is shown with a white line, and borders between 17 HELCOM-defined sub-basins are shown with black lines. BM=Bay of Mecklenburg; GB=Great Belt; KB=Kiel Basin; TS=The Sound.

**Figure 2:** Hypsometry of the Baltic Sea and the HELCOM-defined 17 sub-basins. a) Hypsometry of the entire Baltic Sea, b-r) Hypsometry of HELCOM-defined sub-basins. Hypsometry expressed as accumulated area in percent are shown with red curves for EMODnet and blue for IOWTOPO. Hypsometry expressed as area for 1 m depth intervals are shown with grey filled curves for EMODnet and with purple curves for IOWTOPO.

**Figure 3:** Comparison between mean, median and maximum depths of the HELCOM-defined sub-basins based on the EMODnet (grey bars) and IOWTOPO (blue bars) DBMs. As for Table 1, note that the statistics are calculated from the DBMs' grid cells which have depths representing the mean within the each grid-cell area. This implies, for example, that the maximum depth in each sub-basin will be slightly deeper that shown here. a) Mean depth, b) Median depth, c) Maximum depth.

**Figure 4:** Calculated Terrain Ruggedness Index (TRI) of the Baltic Sea seafloor. Areas specially discussed in the text are shown in the close-up maps a-c.

**Figure 5:** Åland Sea bathymetry, identified sill depths, and bathymetric profiles across the sills based on the EMODnet and IOWTOPO DBMs. a) Bathymetry based on EMODnet. b) Bathymetry based on IOWTOPO. White arrows show the locations of identified bathymetric sills with their depths written next to the arrows. The white stippled lines show the deepest pathways through the area. c) Bathymetric profiles U-U', V-V' and W-W' across the identified sills. Their locations are shown in a and b. Profiles are drawn using both EMODnet and IOWTOPO for comparison. The latter DBM also provides maximum depths of the grid cells. LB=Långskär Basin

**Figure 6:** Bathymetry of the Southern Quark area based on IOWTOPO (a), EMODnet (b) and multibeam bathymetry acquired with RV *Electra* (c). d) Bathymetric profiles X-X' and Y-Y', drawn using the three different bathymetric datasets for comparison, show good correspondence between the RV *Electra multibeam* and EMODnet mean depth, but extremely poor capture by IOWTOPO of ridges and channels relevant to water and sediment mobility.

**Figure 7:** 3D-view of the seafloor in the Southern Quark area based on the multibeam bathymetry acquired with RV *Electra*. a) Overview of the area looking south. b) The 90 m long shipwreck of August Thyssen in ~55 m water depth and an iceberg pit portrayed using a 50×50 cm grid from the multibeam bathymetry. c) Mass wasting, channels carved by bottom current scouring and a potential drift deposit are identified in the bathymetric data. d) Features formed by seafloor interaction with bottom currents.



**Figure 8:** Scatter strength as a function of longitude and depth, from the Kongsberg EK80 split-beam echo sounder (wideband FM mode with centre frequency at 70 kHz). a) the eastern part of the X-X' transect shown in Fig 6c, where examples of fish schools are marked with white arrows and a scattering layer (zooplankton and/or other suspended particles) below 100 m depth marked as black ellipse. b) the Y-Y' transect in Fig 6c, where white ellipse shows example of thermohaline stratification and black ellipse an example of turbulent mixing associated with the steep bathymetry. Note how the stratification (thin horizontal lines within the red ellipse) is interrupted by the turbulence, and is only seen intermittently westward of the steep slope.

Table 1.

| HECLOM sub-basin | EMODnet | | | | | | | IOWTOPO | | | | | |
|---|---|---|---|---|---|---|---|---|---|---|---|---|---|
| | Mean | Median | Max | Stand. Dev. | Area (km²) | Volume (km³) | | Mean | Median | Max | Stand. dev. | Area (km²) | Volume (km³) |
| Kattegat | 22 | 19 | 126 | 16 | 23921 | 532 | | 23 | 20 | 91 | 15 | 22543 | 508 |
| Great Belt | 13 | 12 | 56 | 9 | 10858 | 144 | | 12 | 11 | 38 | 8 | 11733 | 143 |
| The Sound | 12 | 12 | 52 | 7 | 932 | 11 | | 11 | 11 | 32 | 6 | 943 | 11 |
| Kiel Bay | 17 | 18 | 40 | 6 | 3472 | 58 | | 16 | 17 | 30 | 6 | 3475 | 57 |
| Bay of Mecklenburg | 16 | 18 | 31 | 7 | 4613 | 76 | | 16 | 18 | 29 | 7 | 4652 | 75 |
| Arkona Basin | 25 | 21 | 52 | 14 | 17727 | 435 | | 24 | 21 | 50 | 14 | 18191 | 432 |
| Bornholm Basin | 44 | 43 | 100 | 24 | 42150 | 1835 | | 43 | 41 | 95 | 24 | 42638 | 1822 |
| Gdansk Basin | 50 | 49 | 111 | 37 | 5850 | 292 | | 49 | 46 | 113 | 36 | 5833 | 287 |
| Eastern Gotland Basin | 76 | 70 | 243 | 47 | 75019 | 5708 | | 77 | 71 | 241 | 47 | 75132 | 5746 |
| Western Gotland Basin | 73 | 67 | 454 | 51 | 34359 | 2511 | | 68 | 62 | 402 | 50 | 35054 | 2398 |
| Gulf of Riga | 24 | 24 | 66 | 15 | 18705 | 441 | | 22 | 23 | 54 | 14 | 18990 | 423 |
| Northern Baltic Proper | 76 | 74 | 229 | 44 | 32745 | 2496 | | 70 | 69 | 186 | 42 | 33346 | 2348 |
| Gulf of Finland | 37 | 32 | 125 | 26 | 29721 | 1087 | | 34 | 30 | 104 | 23 | 30476 | 1051 |
| Åland Sea | 37 | 19 | 295 | 49 | 16560 | 609 | | 26 | 7 | 243 | 44 | 19494 | 501 |
| Bothnian Sea | 70 | 68 | 288 | 39 | 59326 | 4158 | | 67 | 65 | 265 | 41 | 59925 | 4029 |
| The Quark | 25 | 20 | 122 | 21 | 8287 | 204 | | 19 | 13 | 104 | 19 | 8652 | 164 |
| Bothnian Bay | 43 | 34 | 148 | 32 | 32078 | 1371 | | 39 | 28 | 118 | 32 | 32771 | 1265 |
| **Sum** | | | | | **416320** | **21967** | | | | | | **423848** | **21258** |
| **Baltic Sea (separate calculation)** | 53 | 42 | 454 | 43 | 417115 | 21971 | | 50 | 39 | 402 | 42 | 427470 | 21283 |



Figure 1.



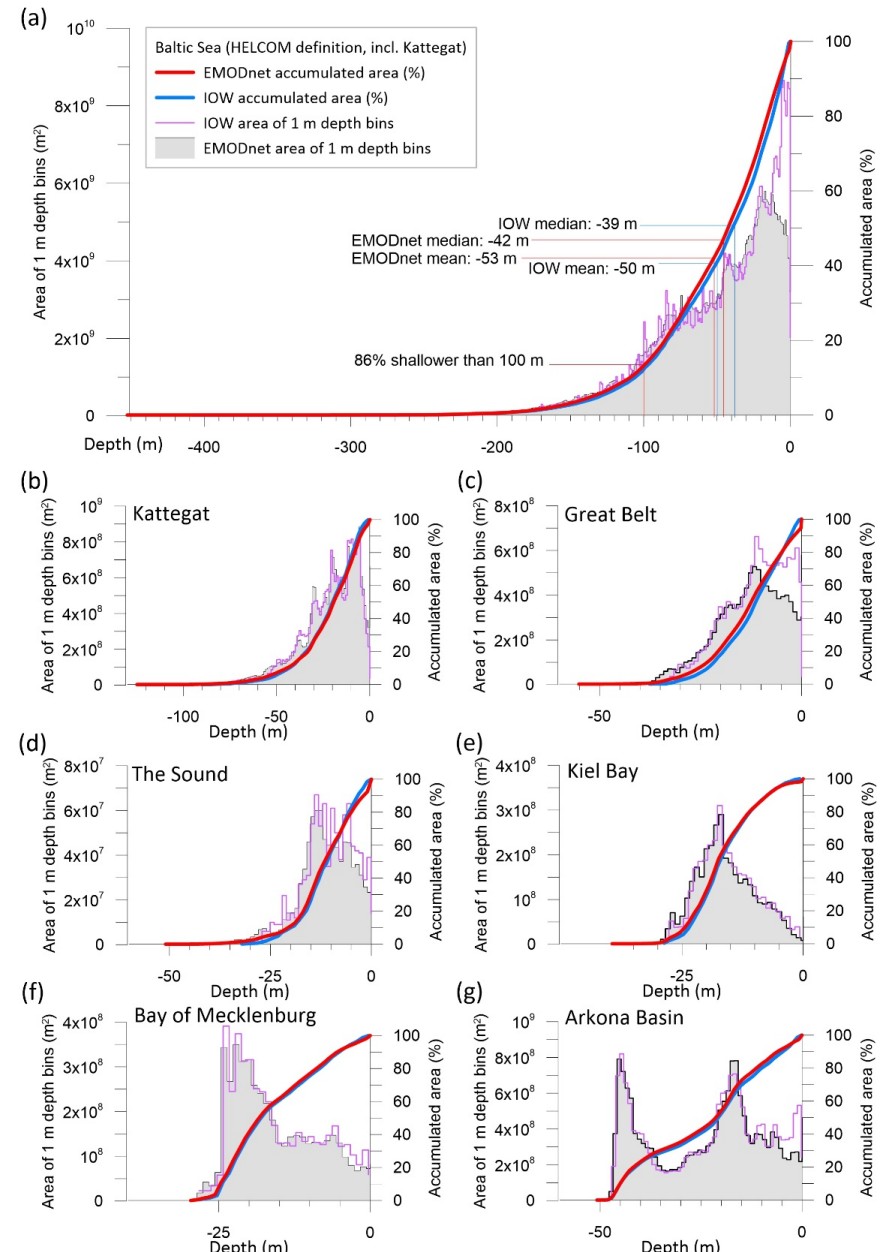

Figure 2a-g





Figure 2h-o



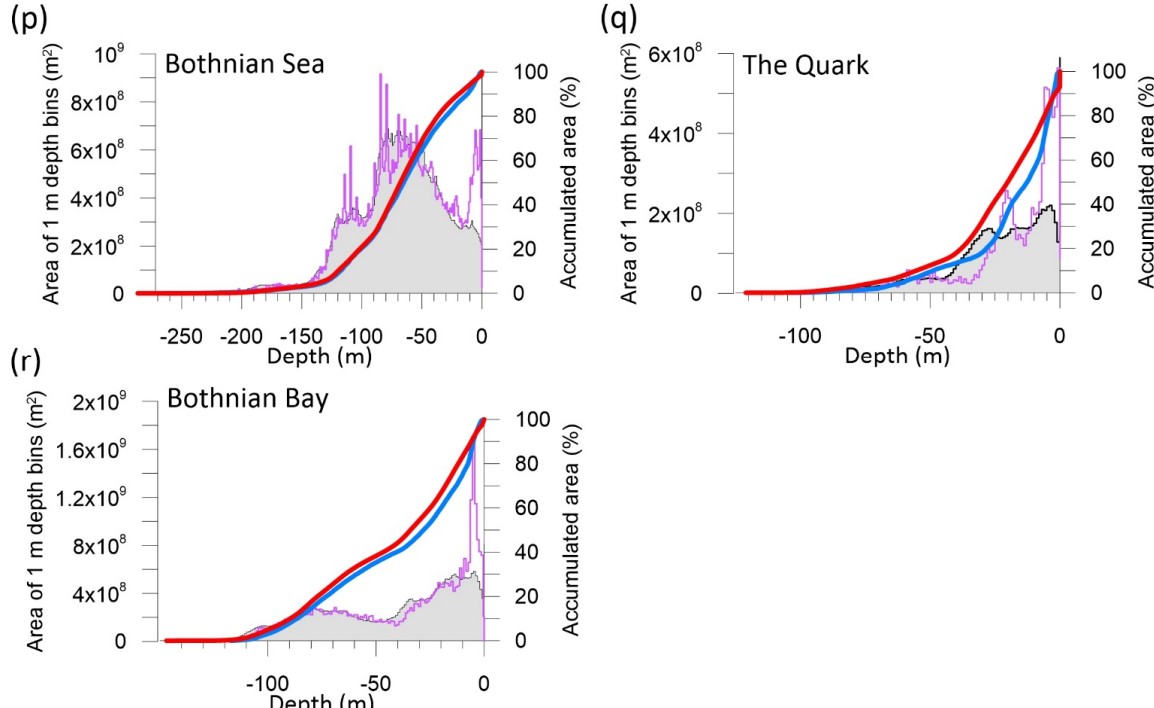

Figure 2p-r





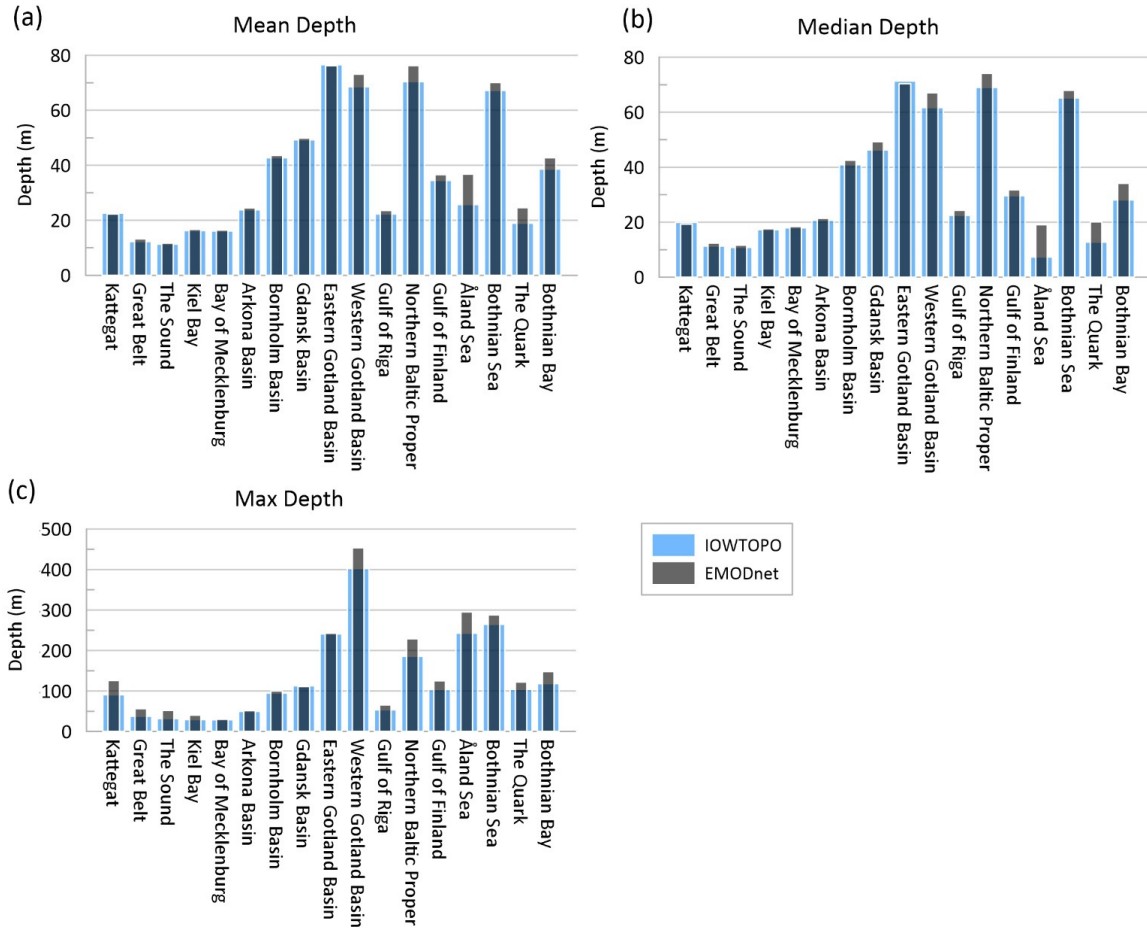

Figure 3

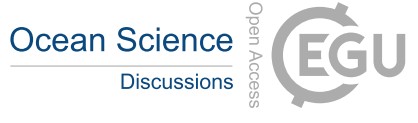



Figure 4


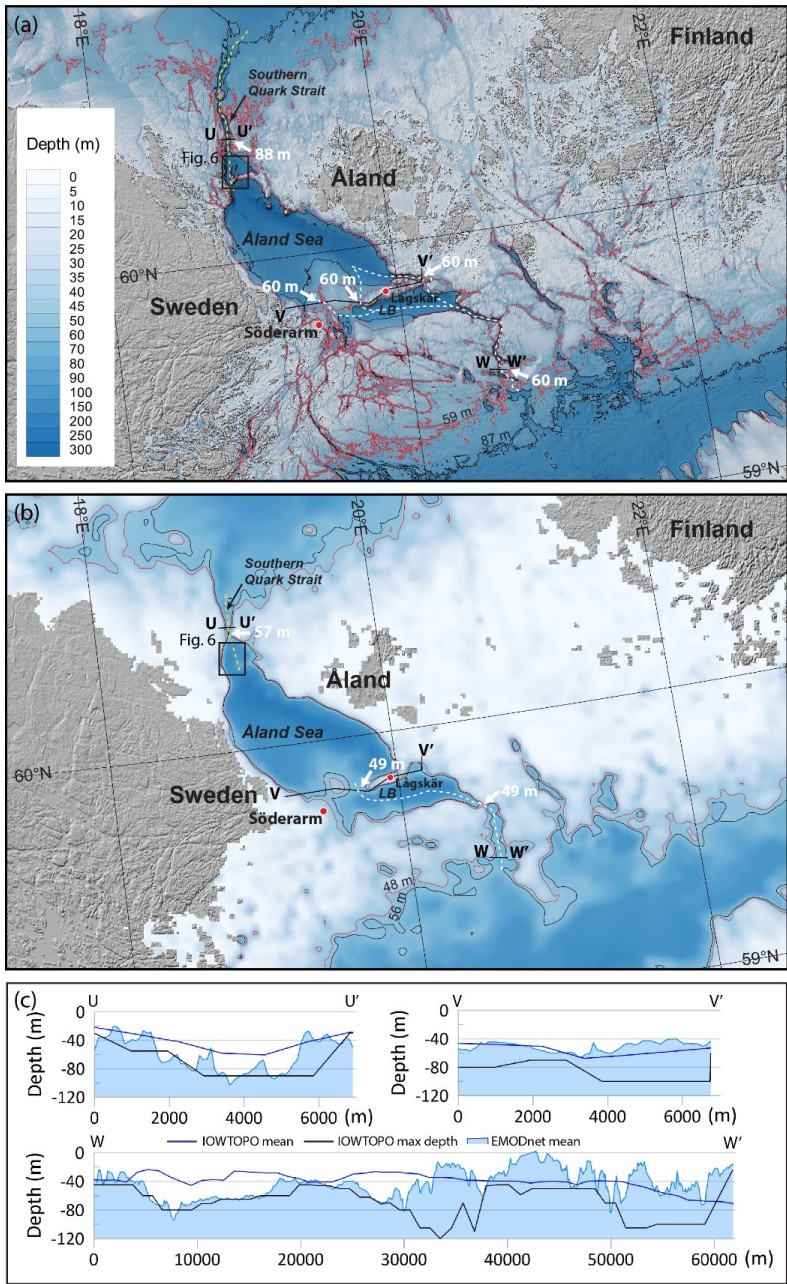

Figure 5




Figure 6



Figure 7



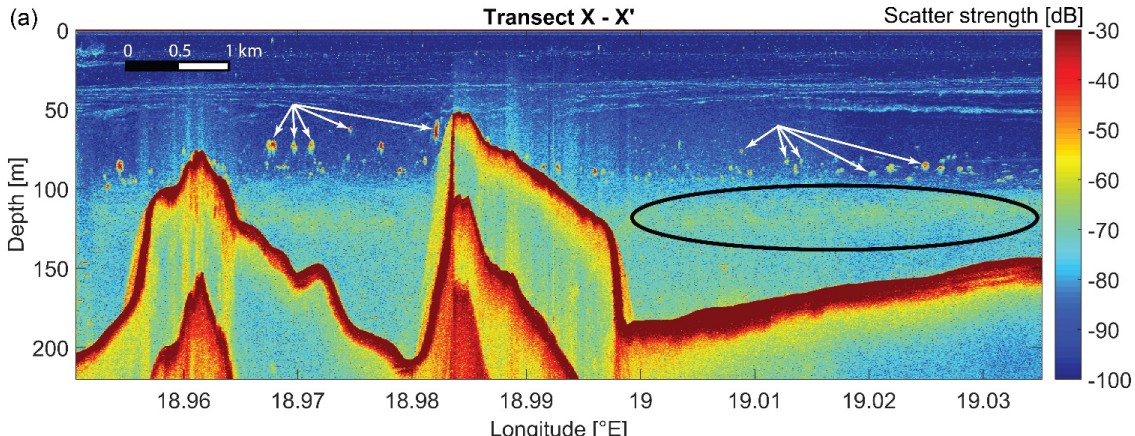

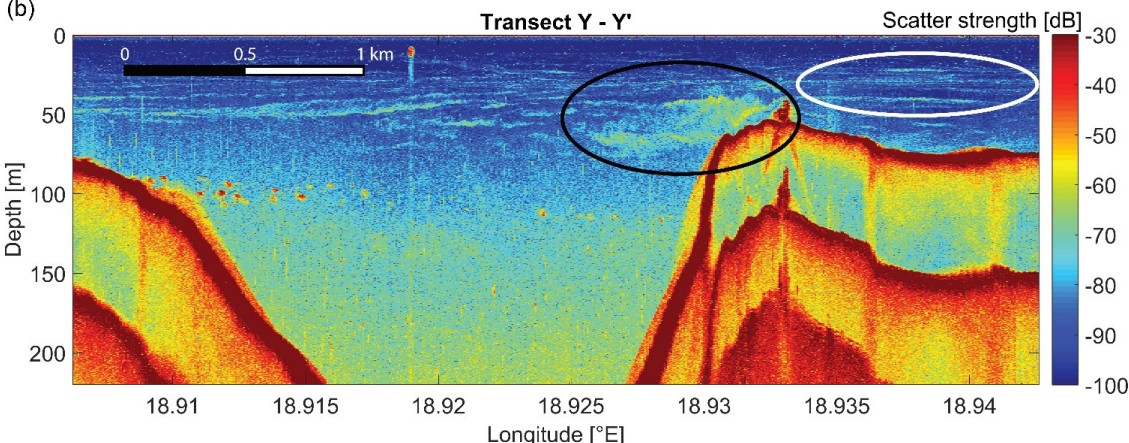

Figure 8