# Peer review of "Bathymetric Properties of the Baltic Sea"

_Ocean Science, 2019_

## Referee Comment (RC1) · Thierry Schmitt (Referee) · 15 Apr 2019

The paper provides a nice overview of recent improvement of the bathymetric knowledge for the Baltic Sea. Recent improvements are principally originating from the European Bathymetric compilation EMODnet Bathymetry. The authors are presenting a comparison between the previous bathymetric Digital Elevation Model, the recent EMODnet Bathymetry and a high resolution recently acquired survey.

Following comments are originating from both the EMODnet community and the Swedish Maritime Administration, which holds some of the bathymetric information in the area:

Page 4 line 15:

"EMODnet DBM is to a large extent based on the same bathymetric source data as the

[Figure]

BSBD"

Comment from me: A lot of data surveyed after 2012 has been added, Better data (300m resolution min and max values in true positions compared to 500m average) from Swedish Territorial waters, Better data delivery from Poland and Latvia, but sadly also a lower data resolution for Denmark.

Page 8 line 18:

"there are no maximum depths provided or, more precisely, those provided are the same as the mean"

Comment from me: This is the effect of that the source data is sparse. We have only been able to use a minimum and maximum values on their true positions filtered by a 300x300m grid due to legal restrictions in the Swedish Territorial sea. For the Finnish part only depths charted in their seacharts has been provided for their territorial area, also due to legal restrictions. All cells having only a mean depth is interpolated and all having the same min, max and mean depth contains a true measured value. In rare occasions both a min and a max value occurs in the same cell giving more trustworthy information about min, max and medium depth.

The resolution of data from the Swedish side of minimum and maximum values within 300m, gives the result that there is only theorethical possibility to populate 2 out of 14 cells.

In the case where the data comes from digitized older, and sparse, surveys (plummets) both the min and max depth in the SMA database (within the selection cell) will be identical and on identical position, hence populating only one of the 14 cells.

In rare cases a min and maximum value from the same or neighbouring selection cells falls within the same Emodnet cell of 115x57.5m (at Lat 60) valid min, max and average values should exist in the EMODNet cell.

In the end this means that even if the area are surveyed by modern methods, only 2

(sometimes 1) of 14 (13.5) emodnet cells contains a measured sounding, the rest is filled by interpolated values.

If instead a cell of 115x115m would have been used the ratio would have been 2 populated cells out of 7 and less north southerly influence on the interpolation of the remaining empty 5 cells.

For the Finnish side the point density is even more sparse than on the Swedish

line 12 redundant use of "using"

Section 2.1. One must know that the fact that "standard deviation" unproperly cal­culated for the EMODnet grid, is limited for the Baltic Sea, and not for the overall EMODnet Grid. You mention the specificity of the data source data locally (section 4.3), please make sure to announce this earlier.

---

## Referee Comment (RC2) · Anonymous Referee #2 · 26 Apr 2019

This paper contains an assessment of Baltic Sea bathymetric properties, comparing results obtained from a older Digital Bathymetric Model (IOWTOPO) and a newly released DBM (EMODnet) and also compares these products to a local very high-resolution multibeam survey. The analysis subdivides the Baltic into several sub-basins to provide regional results.

The authors do a good job at demonstrating the improvements in resolution and accuracy of the newer DBM (EMODnet). They also include a good description of the geospatial pre-processing applied to these data prior to analysis, which is simple for the reader to understand, provides enough information for reproducibility (provided readers have access to the freely available software QGIS) and instills confidence that these data have been treated appropriately. The results, particularly the hyposometry analysis, is well presented and support the interpretation. The paper is logically organised and flows well. The figures are well constructed and well annotated, and the captions

are concise and provide all necessary information. There are no controversial points and the conclusions are clear.

It is pleasing that the authors provide links to all source data. It would be useful if they could also make the TRI map they derived as part of the study available as a gridded product.

Overall, this is a well constructed manuscript and is worthy of publication with minimal revisions. It provides a useful, updated assessment of Baltic Sea bathymetry and highlights the utility of high-resolution multibeam data.

Minor line by line comments/suggestions are provided below.

Page 1:

Line 10: The first sentence could be saved for the introduction and removed from the abstract.

Line 12: "here using the using the" replace with "using the"

Line 22: "we are" replace with "is"

Page 2:

Line 15: (Fig. 1): This is a very nice figure, but personally I find color palettes with more colour variation and/or hill shading more informative.

Line 17: DEM has a resolution of 1/16x1/16 arc minutes. Can you also say what portion is actually sampled at this resolution?

Page 4:

Line 13: "permitting the user to get a view of the source data" replace with "which allows the user to view the source data"

Page 5:

Line 20: "sometime" replace with "sometimes"

Page 7:

Line 9: Perhaps an explanation as to why the differences are largest in the Gulf of Finland and the Quark would be useful or a sentence like "Reasons for these differences are discussed in Section X"

Line 21: "area comes out" replace with "is"

Line 28: "which is as much as" replace with "which is"

Page 8:

lines 9/10: This is a good point. Can you add the section in the Discussion. "This will be addressed further in Section 4.X."

lines 16/17: "Therefore, the threshold..." I found this sentence a little confusing and had to reread it a few times. Could you reword it?

Page 9:

Lines 24/26: "While the wreck..." Again, I found this confusing. I think the word gird needs to be inserted after "2.5 x 2.5 m" on line 25. Also, need to add reference to what the full resolution is. In the caption it states 50 x 50 cm.

Page 10:

Line 26: "EMODnet, which must be considered rather close" replace with "EMODnet. These are close"

Line 31: "will affect the volume by making it smaller" replace with "decrease the total volume"

Page 11:

Line 2: remove word "overall"

Line 11: Reword the start of the ruggedness section. "of geoscientific fields as, for example, heterogeneous seafloor: (1) represents....; (2) has implications.... and (3) influences.."

Page 12:

Line 9: "High TRI values.." replace with "The high TRI values that exhibit a patchy distribution in figure 4c reflect.."

Line 22: "and in past" replace with "and in the past"

Page 13:

Line 25: "so that they lift of the seafloor" should be "lift off? the seafloor". Perhaps reword this.

Page 14:

Line 22: "cases likely even cause" replace with "cases like cause".

Caption for figure 5:

Add full stop after "Basin".

---

## Referee Comment (RC3) · Anonymous Referee #3 · 27 Apr 2019

This manuscript analyses and compares two different digital terrain models of the Baltic Sea: EMODnet and IOWTOPO, and also compares EMODnet to higher-resolution multibeam bathymetric data.

Overall, this research is rigorous and cites appropriate prior published work. It refines some of the basic properties of the Baltic Sea, especially the area and volume, and refines the depths of various sills (important for modeling deep-water exchange between basins). As such, it should become a frequently referenced paper for Baltic Sea bathymetric properties.

The manuscript also identifies some of the weaknesses in the older IOWTOPO, in particular, less source data to enable quantification of minimum and maximum depths in various cells. The hope is that further multibeam bathymetric surveying in the Baltic, as well as other parts of the world, will enable the development of more-statistically

rigourous terrain models in the future.

The authors do identify a research topic worthy of further investigation: how the downgrading of high-resolution terrain models to larger cell sizes can degrade the accuracy of the derived, coarser model, especially where sparse data are used, which is common over much of the world's ocean floor. Development of techniques to more accurately downgrade high-resolution data and models to lower-resolution (larger cell size) will be of significant value to the global bathymetric community, in particular GEBCO.

Side note: there are two section '3.2'. One of page 8, line 13 ("3.2 Bathymetric Sills"), and another on page 9, line 8 ("3.2 High-resolution bathymetry in the Southern Quark").

---

## Author Comment (AC3) · 6 Jun 2019

Reviewer #3 did not require any specific changes, but pointed out that we had mislabeled the sections so that there were two sections 3.2. This has been corrected in the revised version.

---

## Author Response (AR1)

**Summary:** The article "Bathymetric Properties of the Baltic Sea" has been reviewed by three reviewers. We thank the reviewers for their constructive comments. No major concerns are raised and we have addressed all of their comments and implemented their suggested revisions.

**Point-by-point handling of Reviewers' questions and suggested revisions**

**Reviewer #1, Thierry Smith**

Reviewer #1 raises a set of comments with references to specific pages and lines in the manuscript, each comment is here dealt with:

1) Page 4, line 15: Reviewer #1 points out that text here gives the impression of that the EMODnet DBM was largely based on the same data as BSDB, despite that several major updates were made. We propose to revise the key sentence and add a sentence to reflect that EMODnet does indeed provide an update compared to BSDB. It will read:

" The newly released EMODnet DBM is to some extent based on the same bathymetric source data as the BSBD, although new data have been added, specifically in the waters of Poland and Latvia. Furthermore, the input data in Swedish waters were filtered to 300×300 m to meet the nation's legal restrictions, however this provides a more detailed view than the 500×500 m BSDB. All bathymetric source data were compiled on a grid with spherical coordinates at the higher resolution of 1/16×1/16 arc minutes (~115×115 m)."

2) Page 8, line 18: We wrote "there are no maximum depths provided or, more precisely, those provided are the same as the mean" without further explaining the background for this. We agree with the reviewer that this should be commented on, so we propose to add:

"This is an effect of the sparse input data and the down sampling to 300×300 m in Swedish waters where modern multibeam bathymetry exist".

3) We agree with Reviewer #1 that it is important to highlight early that the issues of not providing a standard deviation of the depth is specifically restricted to the Baltic mainly due to the legal restrictions of bathymetric data in Sweden and Finland. We will therefore add in Section 2.1.

"It should be noted that this problem is rather restricted to the Baltic Sea, mainly because other areas of EMODnet do not face the same legal restrictions regarding bathymetric information, specifically an issue in Sweden and Finland, implying that more information have been provided to EMODnet."

**Reviewer #2**

Reviewer #2 does not raise any major concerns, but points out several to the point editorial corrections that we implemented. He/she does suggest that we make available the TRI grid, which we are happy do. We will therefore placed it in the Bolin Centre Database at Stockholm University, along with the appropriate metadata. This will be stated in the revised paper so that the readers can get access to the grid. Reviewer #2 does suggest to either find a color map with more color variations for Figure 1, or add more hill shading. We have decided to add more hill shading rather that changing color map as we do believe that sticking with the blue makes it consistent with the other bathymetric maps in the paper.

Regarding the comment on Page 2, line 17. We are not in the position of providing the exact data coverage, simply because it is not provided to EMODnet due to the restrictions of some countries of providing bathymetry, as now more clearly stated in the paper under section 2.1.

Reviewer #2 agrees with our point raised on page 8 line 9/10 "This highlights the caution needed when interpreting a DBM compiled from heterogeneous source data, something that will be further addressed in the discussion." He/suggests that we add a specific section in the discussion. However, we do follow up on this issue in several places in the discussion. For this reason, we find it redundant to add a specific section and prefer to keep it as is.

**Reviewer #3**

Reviewer #3 did not require any specific changes, but pointed out that we had mislabeled the sections so that there were two sections 3.2. This has been corrected in the revised version.

[revised manuscript text omitted]

Figure 3

[Figure]

Figure 4

[Figure]

Figure 5

[Figure]

Figure 6

[Figure]

Figure 7

[Figure]

[Figure]

Figure 8